# Variable food alters responses of larval crown-of-thorns starfish to ocean warming but not acidification

Benjamin Mos [1,2✉], Naomi Mesic[3] & Symon A. Dworjanyn [3]

Phytoplankton abundance is decreasing and becoming more variable as the ocean climate changes. We examine how low, high, and variable phytoplankton food supply affected the survival, development, and growth of larval crown-of-thorns starfish, *Acanthaster* sp. exposed to combined warming (26, 30 °C) and acidification (pH 8.0, 7.6). Larvae fed a low food ration are smaller, and develop slower and with more abnormalities than larvae fed a high ration. Larvae fed a variable food supply (low, followed by high ration) overcome the negative effects of low food on development rate and occurrence of abnormalities, but are 16–17% smaller than larvae fed the high ration continuously. Acidification (pH 7.6) slows growth and development and increases abnormalities regardless of the food regime. Warming slows growth and development, but these effects are mitigated by high food availability. As tropical oceans warm, the success of crown-of-thorns starfish larvae may depend on the abundance of their phytoplankton prey.

[1] Moreton Bay Research Station (MBRS), School of Biological Sciences, The University of Queensland, 37 Fraser Street, Dunwich, Minjerribah QLD 4183, Australia. [2] Centre for Marine Science (CMS), The University of Queensland, Brisbane, QLD 4072, Australia. [3] National Marine Science Centre, Faculty of Science and Engineering, Southern Cross University, PO Box 4321 Coffs Harbour, NSW 2450, Australia. ✉email: b.mos@uq.edu.au

Marine ecosystems are changing under the influence of a suite of abiotic and biotic stressors[1–3]. Two of the most ubiquitous stressors, warming and acidification, stem from anthropogenic $CO_2$ emissions that increase global temperatures and reduce seawater pH. At present, the surface layers of oceans are on average ~0.75 °C warmer and ~0.1 pH units more acidic than in the middle of the 20th century[2]. By the end of the 21st century, mean sea surface temperatures are likely to increase by an additional 1–4 °C and mean ocean surface pH is likely to drop by a further 0.1–0.3 pH units[2]. Marine ecosystems already experience short periods when increases in temperature and decreases in pH exceed end-of-century predictions, and these periods are becoming more frequent and lasting longer[2,4,5].

In the tropics, ocean warming and associated nutrient limitation are driving an overall decline in phytoplankton abundance[6–9]. This general trend is likely to continue in coming decades as oceans continue to warm[2,9]. At the same time, phytoplankton abundance is expected to become more variable as escalating variability in climatic, seasonal, or weather cycles provide nutrients for short-lived microalgae blooms via upwelling or terrestrial sources[2]. Evidence of increasing variability in the abundance of phytoplankton in tropical systems has already been reported[10,11].

Changes in phytoplankton abundance may have far-reaching consequences in contemporary and future oceans[9]. Phytoplanktons are the primary source of nutrition for the larvae of at least 70% of marine benthic invertebrates[12,13]. Given the direct links between environmental conditions and metabolism in ectotherms, and therefore energy intake, food availability likely exerts a strong influence on the capacity of larval invertebrates to cope with ocean change stressors[14,15]. High food availability can bolster the growth and development of larval echinoderms and molluscs when exposed to low pH or warming that is within thermal thresholds[16–18], whereas low food availability hampers the ability of mollusc larvae to cope with acidification or combined warming and acidification[19–22]. Warming may initially compensate for low food availability by boosting larval development rates, but as larvae age high temperatures tend to exacerbate the negative effects of low food availability[17,18]. Researchers have only recently begun to consider how variation in phytoplankton abundance during the larval period might influence the way in which invertebrates respond to ocean change stressors (e.g., acidification[23]).

This study assessed the response of the larvae of the key coral predator, crown-of-thorns starfish (CoTS, *Acanthaster* sp.) to combined acidification and warming under different food availability regimes. Outbreak populations of *Acanthaster* spp. have caused major coral declines in the Indo-Pacific[24,25]. One of the prevailing explanations posits CoTS outbreaks stem from anthropogenic nutrient influxes that enhance larval survival by fuelling phytoplankton blooms that boost food availability[26,27]. There is a positive relationship between food abundance and growth and development of larval CoTS, except when microalgae densities exceed ~1 × $10^5$ cells m$L^{-1}$ (~10 µg chlorophyll a $L^{-1}$) where greater food availability is detrimental[28–32]. Under optimal food conditions, larval CoTS begin feeding at 2–3 dpf (days post-fertilisation) as bipinnaria and rapidly develop to the late brachiolaria stage by 13–17 dpf after which they are ready to settle and metamorphose into benthic juveniles[28–32]. However, CoTS larvae are also resilient to nutrient-poor conditions, surviving for several weeks in the absence of phytoplankton[29,33].

The influence of phytoplankton abundance on larval CoTS has received little attention in the context of contemporary and future ocean change. CoTS fertilisation and early development are generally unaffected by warming and acidification[34–37] but see ref. [38]. Temperatures above 29 °C can negatively affect the development, growth, and survival of gastrula and later larval stages[35,37,39–41]. Decreases in pH ≥0.2 units slow larval development and can increase mortality, but have disparate effects on larval size[37,38,40,42,43]. Simultaneous acidification and warming can exacerbate the negative impacts of these stressors on gastrulation[38,41] but see refs. [37,40], but appear to have no interactive effects on later bipinnaria and brachiolaria stages[37,38,40]. The way in which larvae respond to acidification and warming may also be influenced by their parents' exposure to these ocean change stressors[37]. All studies that have examined the effects of ocean change stressors on CoTS larvae fed microalgae at rates known to support normal development (0.1–2 × $10^4$ cells m$L^{-1}$). Only one study has examined the potential for different food availabilities to influence resilience to ocean change stressors. When given an abundance of food, larval CoTS grew 30% faster at 30 °C than at 28 °C[18]. No study has examined the influence of food availability on the response of larval CoTS to acidification or combined ocean change stressors.

To address these knowledge gaps, we tested the effects of (1) low food availability, (2) low followed by high food availability, and (3) high food availability on the responses of CoTS larvae to present-day (pH 8.0, 26 °C) and acidification (pH 7.6) and warming (30 °C) scenarios. We expected larvae fed a high food ration to be more resilient to acidification and warming than larvae fed a low food ration. We further hypothesised that in present-day conditions, the growth and development of CoTS would be slowed by a short period of low food and recover when food availability was increased, but overall survival would be unaffected. However, in warmed and acidified conditions, CoTS initially fed a low food ration for 1 week and then a high food ration thereafter were expected to have similar development or survival compared to larvae fed a constant low food ration due to the combined impacts of carry-over effects of poor nutrition and increased energy demand for maintaining homoeostasis and growth in stressful environments[44].

We found larval CoTS fed a low food ration developed and grew slower and had more abnormalities than larvae fed a high food ration. Larvae quickly overcame some of the negative effects of low food when switched from a low to a high food ration. Exposure to warming (30 °C) slowed development and growth and increased the occurrence of abnormal larvae. High food availability helped larvae cope with warming (30 °C), with some larvae fed the variable and high food rations completing larval development by 40 dpf, while all larvae in the warming conditions fed a low food ration experienced arrested development and died by week 6. The food regime had little effect on the response of larvae to acidification. Larvae held in acidified conditions (pH 7.6) grew and developed slower and experienced an increased occurrence of abnormalities compared to larvae in ambient conditions (pH 8.0) regardless of the food regime. Our results indicate that the abundance of phytoplankton food can influence whether CoTS exposed to high temperatures complete larval development, but there appears to be little influence of food on the way in which larvae respond to acidification at the levels tested here.

## Results

**Survival**. Survival was followed to 56 dpf (days post-fertilisation) when larvae in each replicate had reached the experimental endpoint (>50% of living larvae at the late brachiolaria stage) or all larvae had died (Supplementary Fig. 1). Larvae in ambient treatments (pH 8.0, 26 °C) reached the late brachiolaria stage by week 4, therefore survival was not formally analysed beyond this time point (Fig. 1, Supplementary Table 1). Survival during weeks 1–3 was not influenced by food but was affected by a significant

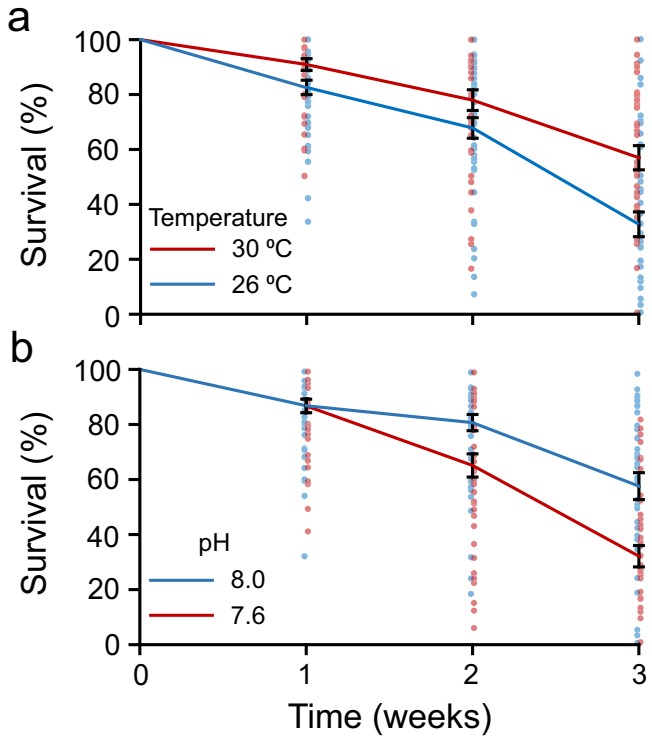

**Fig. 1 Effects of temperature and pH on survival of *Acanthaster* sp. larvae.** Larvae were reared in all combinations of two temperatures (26, 30 °C), two pH levels (pH 8.0, 7.6), and three food level treatments (low, switch, high) in a flow-through seawater system. Larvae in present-day treatments (pH 8.0, 26 °C) reached the late brachiolaria stage and began to settle by week 3, so survival was not formally analysed beyond this time point. Survival depended on a significant temperature × weeks interaction and a significant pH × weeks interaction ($p < 0.05$), with food treatment and all other interactions having little effect (repeated measures ANOVA, Supplementary Table 1). For ease of illustration, data for the **a** temperature × weeks interaction and **b** pH × weeks interaction are presented here, and the full data set is presented in Supplementary Fig. 1. At 3 weeks, survival was significantly greater for larvae reared at 30 °C than 26 °C, and greater for larvae grown at pH 8.0 than pH 7.6 (Supplementary Table 1). Survival was calculated using data collected for all living larvae regardless of their morphology (i.e., included abnormal and normal larvae). Solid lines represent the mean for each treatment. Error bars (black) represent the standard error of the mean. For all treatments, $n = 42$.

interaction between temperature and time (Fig. 1a, Supplementary Table 1). There were 1.1 and 1.7 times more larvae surviving in the 30 °C treatment than in the 26 °C treatment at 1 and 3 weeks respectively (Fig. 1a, Supplementary Table 1). Survival was not different between temperature treatments at 2 weeks post-fertilisation (Fig. 1a, Supplementary Table 1). Survival during weeks 1–3 also depended on a significant interaction between pH and time (Fig. 1b, Supplementary Table 1). After 1 week, there was no difference in survival between pH 8.0 and pH 7.6 treatments (Fig. 1b, Supplementary Table 1), but there were 19% and 44% fewer larvae in pH 7.6 treatments than in pH 8.0 treatments at 2 and 3 weeks post-fertilisation respectively.

**Larval size**. Growth of larval CoTS was influenced by the temperature and pH of their culture water, and the amount of microalgae they were fed (Fig. 2a, b, Supplementary Table 1, Supplementary Fig. 2, Supplementary Table 2). At 11 dpf, larval length depended on a significant interaction between temperature and pH and was also affected by food independently of

temperature and pH (Fig. 2a, Supplementary Table 1). In the 26 °C treatment, larvae reared at pH 7.6 were 25% shorter than larvae at pH 8.0 (Fig. 2a, Supplementary Table 1). In the 30 °C treatment, larvae reared at pH 8.0 and pH 7.6 did not differ in length (Fig. 2a, Table 1). At pH 8.0, larvae reared at 30 °C were ~25% shorter than larvae at 26 °C (Fig. 2a, Supplementary Table 1). At pH 7.6, larvae reared at 26 and 30 °C did not differ in length. Larvae fed $1 \times 10^3$ cells mL$^{-1}$ (low and switch food treatments) were ~23% shorter than larvae fed $5 \times 10^4$ cells mL$^{-1}$ (high food treatment) (Fig. 2a, Supplementary Table 1). There was no significant difference in the length of larvae fed the low and switch food treatments (Fig. 2a, Supplementary Table 1).

At 11 dpf, larval width depended on a significant interaction between temperature and pH and was also affected by food independently of temperature and pH (Fig. 2b, Supplementary Table 1). In the 26 °C treatment, larvae reared at pH 7.6 were 21% narrower than larvae reared at pH 8.0 (Fig. 2b, Supplementary Table 1). In the 30 °C treatment, larvae reared in pH 8.0 and pH 7.6 treatments did not differ in width. At pH 8.0, larvae reared at 30 °C were ~24% narrower than larvae at 26 °C (Fig. 2b, Supplementary Table 1). At pH 7.6, larvae reared at 26 and 30 °C did not differ in width. Larvae fed $1 \times 10^3$ cells mL$^{-1}$ (low and switch food treatments) were ~20% and ~18% narrower than larvae fed $5 \times 10^4$ cells mL$^{-1}$ (high food treatment) respectively (Fig. 2b, Supplementary Table 1). There was no significant difference in the width of larvae fed the low and switch food treatments (Fig. 2b, Supplementary Table 1).

At 18 dpf, larval length was significantly affected by temperature, pH, and food, with no interactions among these factors (Fig. 2a, Supplementary Table 1). Larvae reared at 30 °C were ~28% shorter than larvae at 26 °C (Fig. 2a, Supplementary Table 1). Larvae reared at pH 7.6 were ~32% shorter than larvae at pH 8.0 (Fig. 2a, Supplementary Table 1). Larvae fed the low and switch food treatments were ~35% and ~16% shorter than larvae fed the high food treatment respectively (Fig. 2a, Supplementary Table 1). Larvae fed the low food treatment were ~22% shorter than larvae fed the switch food treatment (Fig. 2a, Supplementary Table 1).

At 18 dpf, larval width was significantly affected by temperature, pH, and food, with no interaction among these factors (Fig. 2b, Supplementary Table 1). Larvae reared at 30 °C were ~29% narrower than larvae at 26 °C (Fig. 2b, Supplementary Table 1). Larvae reared at pH 7.6 were ~29% narrower than larvae reared at pH 8.0 (Fig. 2b, Supplementary Table 1). Larvae fed the low and switch food treatments were ~33% and ~17% narrower than larvae fed the high food treatment respectively (Fig. 2b, Supplementary Table 1). Larvae fed the low food treatment were ~19% narrower than larvae fed the switch food treatment (Fig. 2b, Supplementary Table 1).

**Abnormal larvae**. At 11 dpf, the proportion of abnormal larvae present (%) depended on a significant interaction between temperature and food, but there was no effect of pH and no interactions between pH and other factors (Fig. 3a, Supplementary Table 1). In replicates fed the low and switch food treatments (i.e., $1 \times 10^3$ cells mL$^{-1}$), there were ~3.2 and ~1.7 times more abnormal larvae present at 30 °C than at 26 °C respectively (Fig. 3a, Supplementary Table 1). In replicates fed the high food treatment, there was no significant difference in the proportion of abnormal larvae at 26 and 30 °C (Fig. 3a, Supplementary Table 1). At 26 °C, there was no significant difference in the proportion of abnormal larvae present in replicates fed low, switch, or high food treatments (Fig. 3a, Supplementary Table 1). At 30 °C, there were ~1.9 times more abnormal larvae present in replicates fed the low food treatment than in replicates fed the high food treatment

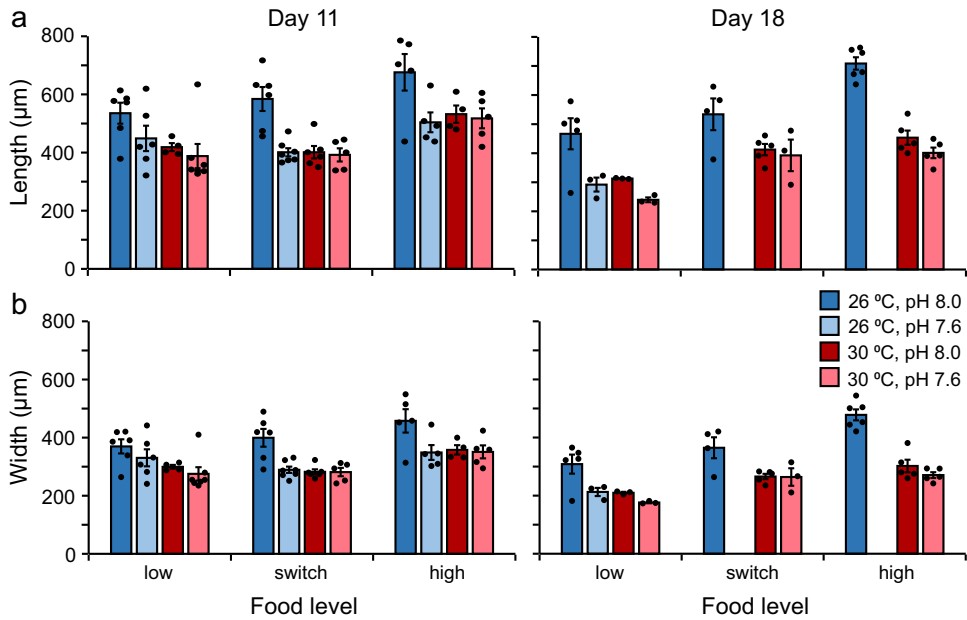

**Fig. 2 Effect of temperature, pH, and food on the size of *Acanthaster* sp. larvae. a** Length. **b** Width. Larvae were grown in all combinations of two temperatures (26, 30 °C), two pH levels (pH 8.0, 7.6), and three food level treatments (low, switch, high) in a flow-through seawater system. Larvae were fed *Proteomonas sulcata* three times per day at a rate equivalent to $1 \times 10^3$ cells mL$^{-1}$ (low), $1 \times 10^3$ cells mL$^{-1}$ from 3 to 11 dpf (days post-fertilisation) and $5 \times 10^4$ cells mL$^{-1}$ thereafter (switch), or $5 \times 10^4$ cells mL$^{-1}$ (high). Bars represent means. Error bars represent the standard error of the mean. Black dots represent replicates. Measurements of length and width were not made on larvae that displayed abnormal morphology (see Fig. 3). Data on length and width for the 26-7.6-switch and 26-7.6-high treatments at 18 dpf were not presented due to low replication ($n < 3$) because of high mortality (Supplementary Fig. 1) and a high occurrence of larvae with abnormal morphology in these treatments.

**Table 1 Outcome of binary logistic regression.**

| Predictor | Comparison | Wald $\chi^2$ | p | b | Lower | Odds | Upper |
|---|---|---|---|---|---|---|---|
| **Temperature** | **26 vs. 30** | **6.01** | **0.014** | **−5.00 (−25.29, −1.02)** | **1.23E-5** | **0.007** | **0.37** |
| **pH** | **8.0 vs. 7.6** | **6.47** | **0.011** | **−5.49 (−42.90, 9.82)** | **6.00E-5** | **0.004** | **0.28** |
| **Food level** | | **7.18** | **0.028** | | | | |
| | **high vs. low** | **7.18** | **0.007** | **−2.66 (−20.83, −1.72)** | **0.01** | **0.07** | **0.49** |
| | high vs. switch | 1.07 | 0.300 | | | | |
| Temperature × pH | | 3.27 | 0.071 | | | | |
| **Constant** | | **8.61** | **0.003** | | | | |

Coefficients of the model predicting whether any *Acanthaster* sp. larvae within a replicate would develop morphological structures indicative of their capacity to undergo settlement and metamorphosis (i.e., become competent) in 12 temperature-pH-food treatments. Confidence intervals (95% bias-corrected and accelerated) are based on 1000 bootstrap samples. The bold type indicates predictors with significant differences among levels according to Wald $\chi^2$ tests ($p < 0.05$). Note $R^2 = 0.31$ (Hosmer–Lemeshow), 0.28 (Cox–Snell), 0.42 (Nagelkerke). Model $\chi^2$ (5) = 27.27, $p < 0.00005$. Overall percentage correct = 84.5%.

(Fig. 3a, Supplementary Table 1). The proportion of abnormal larvae present in replicates fed the switch food treatment was not significantly different than replicates fed the low or high food treatments (Fig. 3a, Supplementary Table 1).

At 18 dpf, the proportion of abnormal larvae present ranged from 7.1 to 79.4% and was highly variable within and among treatments (Fig. 3b). The proportion of abnormal larvae present depended on a significant three-way interaction between temperature, pH, and food (Fig. 3b, Supplementary Table 1). Post-hoc pair-wise tests revealed complex outcomes depending on the combination of temperature, pH, and food treatments, with no clear patterns (Supplementary Table 1).

**Larval development**. At 11 dpf, the proportion of the most developed larval stage (early brachiolaria) in replicates was significantly affected by temperature, pH, and food, with no interaction among these factors (Fig. 4a, Supplementary Table 1). There were ~2.1 times more early brachiolaria present in 26 °C treatments than in 30 °C treatments (Fig. 4a, Supplementary

Table 1). There were ~2.6 times more early brachiolaria in pH 8.0 treatments than in pH 7.6 treatments (Fig. 4a, Supplementary Table 1). There were ~3.8 and ~4.9 times more early brachiolaria in replicates fed $5 \times 10^4$ cells mL$^{-1}$ (high food treatment) than those fed $1 \times 10^3$ cells mL$^{-1}$ (low and switch food treatments) respectively (Fig. 4a, Supplementary Table 1). There was no significant difference in the proportion of brachiolaria in replicates fed low and switch food treatments (Fig. 4a, Supplementary Table 1).

At 18 dpf, data on larval stages for pH 7.6 treatments had low replication ($n < 3$) due to poor survival (Supplementary Fig. 1) and a high occurrence of abnormal larvae (Fig. 3b). At 18 dpf, the proportion of the most developed larval stage (late brachiolaria) in the pH 8.0 treatments depended on a significant interaction between temperature and food (Fig. 4b, Supplementary Table 1). At 26 °C, downwellers fed the high food treatment had ~3.0 times more late brachiolaria than those fed the low food treatment (Fig. 4b, Supplementary Table 1). The proportion of late brachiolaria in replicates fed the switch food treatment did not

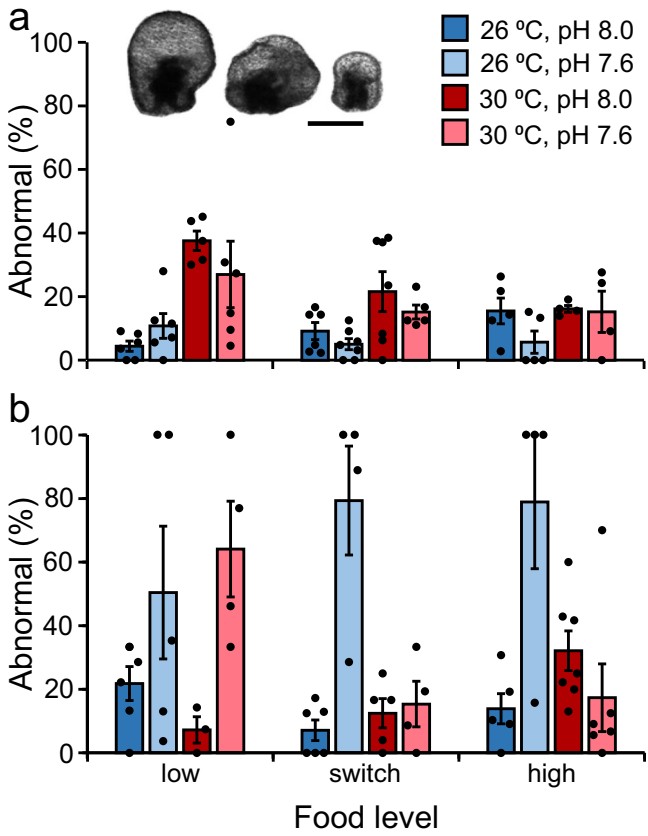

**Fig. 3 Effect of temperature, pH, and food on the occurrence of abnormal *Acanthaster* sp. larvae.** The proportion of *Acanthaster* sp. larvae that were abnormal when reared in two temperatures (26, 30 °C), two pH levels (pH 8.0, 7.6), and three food level treatments (low, switch, high) in a flow-through seawater system. **a** 11 dpf (days post-fertilisation). **b** 18 dpf. Larvae were fed *Proteomonas sulcata* three times per day at a rate equivalent to $1 \times 10^3$ cells mL$^{-1}$ (low), $1 \times 10^3$ cells mL$^{-1}$ from 3 to 11 dpf and $5 \times 10^4$ cells mL$^{-1}$ thereafter (switch), or $5 \times 10^4$ cells mL$^{-1}$ (high). Bars represent means. Error bars represent the standard error of the mean. Black dots represent replicates. Inset: Examples of abnormal larval morphologies. Scale bar = 100 μm. Photo credit: N. Mesic. Examples of normal larval morphology are provided in Fig. 4.

differ from replicates fed the low or high food treatments (Fig. 4b, Supplementary Table 1). At 30 °C, there was no difference in the proportion of late brachiolaria in replicates fed low, switch, or high food treatments (Fig. 4b, Supplementary Table 1).

**Competence to settle**. Late brachiolaria stage larvae with a prominent rudiment indicative of competence to undergo settlement and metamorphosis to post-larval stages were seen in some treatments from 16 dpf (Fig. 5a, b). At the end of the experiment, the treatment with the greatest number of replicates that had competent larvae was the ambient control treatment (26 °C, pH 8.0) fed the high food treatment, where larvae attained competence in six of seven replicates at 16 dpf (Fig. 5a, b; mean total surviving larvae per replicate = 1108 ± 83 SE, n = 6, of which 54.9% ± 6.8 SE were late brachiolaria at 16 dpf). When compared to the high food/26 °C/pH 8.0 treatment, treatments that were acidified (pH 7.6), warmed (30 °C), and had reduced food rationing (low and switch food levels) had fewer replicates with competent larvae (<4 of 7, Fig. 5a;) and larvae took as much as 2.4 times longer to obtain competence (Fig. 5b; range total surviving

larvae per replicate = 9–1022 at 28–35 dpf). In 5 of 12 treatments, all larvae died without becoming competent to settle (Fig. 5a, b).

A binary logistic regression analysis comparing the effects of temperature, pH, and food level on whether any larvae in a replicate would attain competence to settle was statistically significant $\chi^2$ (5) = 27.27, $p < 0.00005$ (Table 1). The model explained 42% (Nagelkerke $R^2$) of the variation in whether any larvae in a replicate attained competency to settle and correctly classified 84.5% of cases. Competent larvae were ~143 times less likely to be in replicates maintained at 30 °C than in replicates at 26 °C (Table 1, Fig. 5a). Similarly, competent larvae were 250 times less likely to be in replicates with seawater at pH 7.6 than in replicates with ambient pH (pH 8.0), although this should be interpreted with caution as the confidence interval for *b* crossed zero (Table 1, Fig. 5a). Food also had a significant influence on the likelihood that larvae attained competence to settle (Table 1, Fig. 5a). Competent larvae were 100 times less likely to be in replicates fed the low food treatment compared to the high food treatment (Table 1). However, competent larvae were equally as likely to be in replicates fed the switch food treatment as those fed the high food treatment (Table 1). The interaction between pH and temperature was included in the best-fitting model but was not significant ($p < 0.071$, Table 1). All other interactions tested did not contribute to improving the model.

## Discussion

CoTS larvae fed a high food ration ($5 \times 10^4$ cells mL$^{-1}$ *Proteomonas sulcata*, three times a day) grew and developed faster than larvae fed a low food ration ($1 \times 10^3$ cells mL$^{-1}$, three times a day). This was unsurprising given the well-documented positive relationship between phytoplankton abundance and growth of CoTS fed at levels similar to the current study[28–32]. The relationship between phytoplankton abundance and the size of CoTS larvae varies among studies and this variation appears to have been generated by differences in experimental methods[28–32]. For instance, in this study where flow-through water was provided, larvae fed the high food ration were ~30–50% larger compared to similarly-aged CoTS fed a comparable concentration of the same species of microalgae in static culture conditions[28,29], but were generally smaller than larvae fed a constant supply of two species of microalgae at rates of ~0.8–1.1 × 10³ cells mL$^{-1}$ in a flow-through system[32]. Larger larvae generally have better post-larval fitness[45,46]. However, because CoTS and many other echinoderm larvae can increase the size of their feeding structures when food is limited[47], starved or underfed larvae can be as large as well-fed individuals but are unlikely to survive to the juvenile stage[28]. The potential for larval fitness to be influenced by experimental setup and the way in which microalgae are fed (pulse vs. constant) warrants investigation to better understand how well food rations identified as optimal in the laboratory correspond with phytoplankton abundances that enhance the success of larval CoTS in the wild.

A high food ration helped CoTS to cope with a 4 °C increase in temperature. At 30 °C, some larvae fed a high food ration were ready to settle in as few as 18 days, whereas all of the larvae fed a low food ration did not reach the late brachiolaria stage and died by week 6. Previous studies on CoTS and the purple sea star, *Pisaster ochraceus*, also found larvae exposed to temperatures near their upper thermal limit developed normally when fed a high food ration but died in the same conditions when food was limited[18,48]. Higher temperatures within thermal limits will increase the basal metabolic rate of ectothermic larvae but low food rations may provide insufficient energy in warming treatments to keep the aerobic scope of the larvae in the positive, resulting in death[44,49]. Alternatively, the physiological

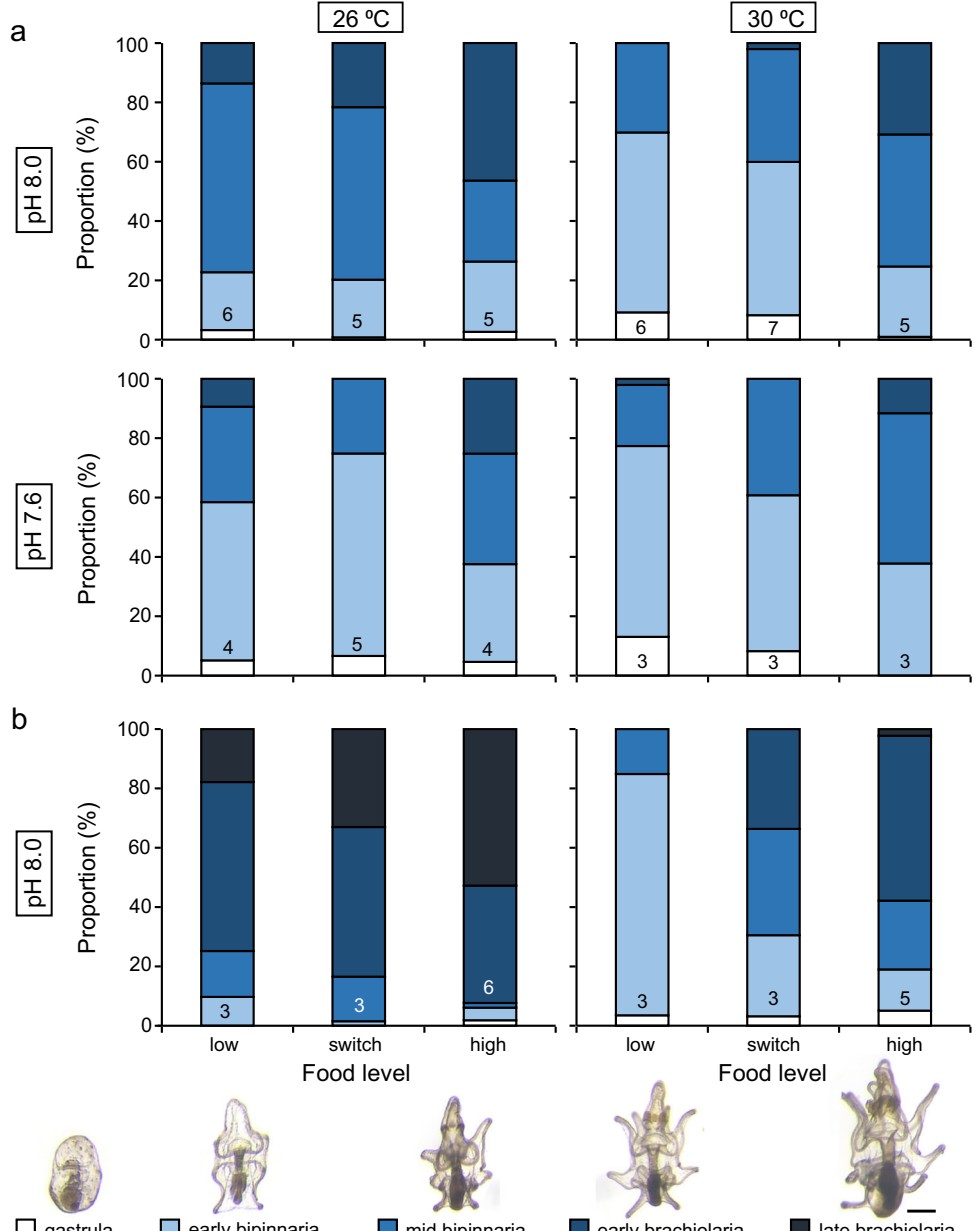

**Fig. 4 Effect of temperature, pH, and food on the development of *Acanthaster* sp. larvae.** The proportion of larval stages (%) of *Acanthaster* sp. larvae present when reared at two temperatures (26, 30 °C), three food level treatments (low, switch, high), and two pH levels (pH 8.0, 7.6) for **a** 11 dpf (days post-fertilisation) and **b** one pH level (pH 8.0) for 18 dpf, in a flow-through seawater system. Larvae were fed *Proteomonas sulcata* three times per day at a rate equivalent to $1 \times 10^3$ cells mL$^{-1}$ (low), $1 \times 10^3$ cells mL$^{-1}$ from 3 to 11 dpf and $5 \times 10^4$ cells mL$^{-1}$ thereafter (switch), or $5 \times 10^4$ cells mL$^{-1}$ (high). Data on larval stages at 18 dpf for pH 7.6 treatments were not presented due to low replication ($n < 3$) because of low survival (Supplementary Fig. 1) and a high occurrence of abnormal larvae in these treatments. Data are means. The number at the base of each bar is the number of replicates for which data was available. Legend: Larval stages appear in scale relative to each other, scale bar = 200 µm. Photo credits: N. Mesic.

mechanisms used by larvae to counter the stress of inadequate food intake might also reduce their capacity to cope with increases in temperature[49]. The results of this study and Uthicke et al.[18] support our hypothesis that CoTS larvae exposed to high temperatures differ in their responses when fed different amounts of food, though these effects appear to only become obvious late in larval development.

Low food abundance slowed development and growth, but CoTS larvae rapidly recovered when switched to a high food ration. Larvae fed $1 \times 10^3$ cells mL$^{-1}$ until 11 dpf and then $5 \times 10^4$ cells mL$^{-1}$ thereafter were significantly larger and more developed at 18 dpf compared to larvae fed the low food ration only

and were just as likely to reach the late brachiolaria stage as larvae fed a high food ration continuously. This is the first study to test the effects of variable food conditions on larval CoTS, though similar experiments have been done using a diverse suite of invertebrates and vertebrates (e.g., refs. [50–57]). A common finding is that larvae display compensatory growth when food limitation is reversed, completing larval development in a similar amount of time as consistently well-fed larvae; but there are often negative carry-over effects that can extend into adulthood[55,57]. There was evidence of carry-over effects of food limitation for CoTS in this study. Larvae fed the switch ration were 16–17% smaller at 18 dpf than larvae fed the high food ration continuously. The

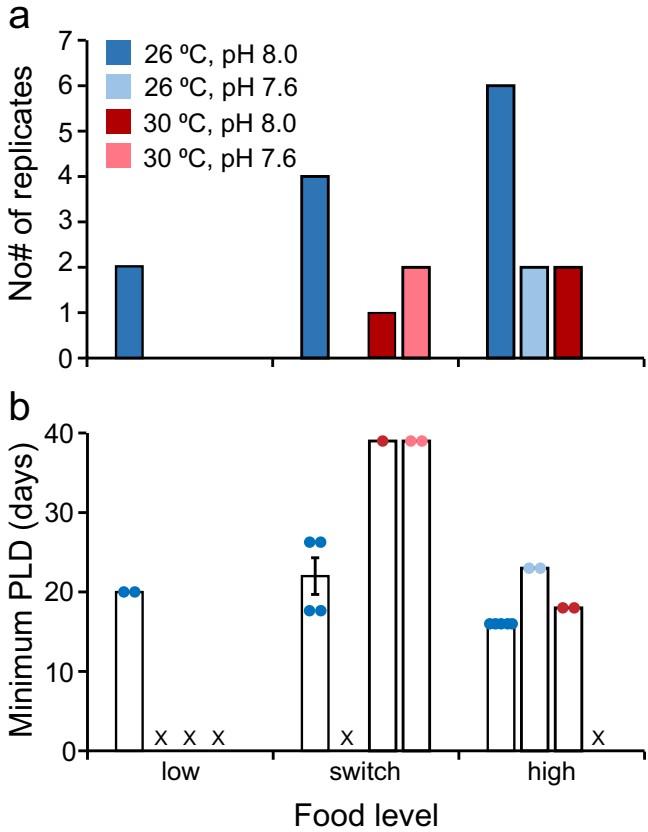

**Fig. 5 Effect of temperature, pH, and food on the competence of *Acanthaster* sp. larvae.** Larvae were reared in all combinations of two temperatures (26, 30 °C), two pH levels (pH 8.0, 7.6), and three food level treatments (low, switch, high) in a flow-through seawater system. **a** Number of replicates in each treatment (out of seven total) where at least one larva developed to the late brachiolaria stage and had a large rudiment indicative of competency to undergo settlement and metamorphosis to post-larval stages. **b** Minimum larval pelagic duration (PLD); the number of days post-fertilisation (dpf) elapsed before any competent larvae were sighted. X identifies treatments where all larvae died before attaining competence to settle. Bars represent means. Error bars represent the standard error of the mean. The number of replicates in each treatment is shown by the data points and corresponds with the values shown in (**a**).

implications of smaller larvae in CoTS are not clear, but there is a large body of evidence that suggests smaller larvae have poor post-larval fitness[45,46].

We found evidence that exposure to warming impedes the recovery of larvae after a short period of food limitation. At 26 °C, larvae fed the switch ration took a similar amount of time to reach the late brachiolaria stage as larvae fed the high ration, but at 30 °C, larvae fed the switch ration took more than twice as long as larvae fed the high ration to complete larval development. One implication of this finding is that CoTS exposed to high temperatures after a period of food limitation are likely to face a higher risk of mortality due to an extended larval period. The risk of predation, offshore transport, and other sources of mortality increases with duration in the plankton[12,58]. The only other study to our knowledge to test the effects of any ocean change stressor on the recovery of food-limited marine larvae found little effect of elevated $p$CO$_2$ on mortality, growth, and lipid content of 9-day-old larvae of two species of oyster following 5 days starvation[23]. We did not find any temperature × food regime interactions in the first 18 days, except for the occurrence of abnormalities, suggesting that the effects of high temperature on recovery from

food limitation may take time to become apparent. A potential explanation for why we found abnormal larvae were most prevalent in high temperature (30 °C), low food level, and/or low pH treatments may be because small clones were classified as arrested or abnormal larvae. Many larval asteroids including CoTS can create clones within days after fertilisation[48,59,60], and exposure to stressful conditions such as increased temperature promotes cloning in larval echinoderms[48,61]. It is important to note, however, all CoTS larvae held at 30 °C died when fed a low food ration, and switching to a high food ration meant some larvae reached the final larval stage. Thus, despite the potential for warming to slow recovery and exacerbate negative carry-over effects, encountering high food abundance after a period of food limitation is likely to enhance larval success in warming scenarios.

A high food ration was not helpful for CoTS larvae exposed to low pH (7.6), and there was no significant food × pH interaction for survival, growth, or development. In other studies, primarily on molluscs, high food availability increased the capacity of marine invertebrates to cope with reduced pH, likely by providing additional resources to meet the increased metabolic demands of maintaining homoeostasis and calcification in stressful conditions[14,62]. One explanation for why we found no evidence of a food × pH interaction is that exposure to acidification conditions can delay the onset of feeding and decrease the ingestion rates of marine larvae[19,63]. It may be that the ingestion rates of larval CoTS exposed to low pH conditions were depressed regardless of food availability. Ingestion in echinoderm larvae appears particularly sensitive to low pH, although the mechanism through which low pH limits feeding rates has yet to be identified[64]. Exposure to low pH may also impede digestion in echinoderm larvae[65,66].

Today and in the future, tropical marine larvae are more likely to encounter heat waves, warmer temperatures, and more highly variable food conditions than during previous centuries[2]. Short-lived phytoplankton blooms can occur simultaneously with marine heat waves, stimulated by the release of nutrients from dead organisms[67], but more often occur in the days to weeks after heat waves because of coastal upwellings that bring nutrient-rich water to the surface[68,69]. At low and mid-latitudes, the most extreme marine heat waves tend to coincide with reduced chlorophyll *a* concentrations that are indicative of low phytoplankton abundances[70,71]. Short-lived heat waves and phytoplankton blooms occur within a broader context of increasing average ocean temperatures and declining phytoplankton abundance[2,9]. Our results indicate low phytoplankton abundances are likely to exacerbate the impacts of ocean warming and marine heat waves on invertebrate larvae. However, the larvae of some species such as CoTS may be able to take advantage of phytoplankton blooms to better cope with or recover from the negative impacts of high temperatures. This study highlights the success of CoTS larvae in contemporary and future oceans depends on the abundance and variability of its phytoplankton prey, indicating that previous ocean change studies on this species should be re-evaluated with this context in mind.

## Methods

**Experimental setup.** Experimental treatments consisted of two temperatures (26, 30 °C), two pH (pH 8.0, 7.6) and three food levels ('low': $1 \times 10^3$ cells mL$^{-1}$, 'switch': $1 \times 10^3$ cells mL$^{-1}$ from 3 to 11 dpf, followed by $5 \times 10^4$ cells mL$^{-1}$ thereafter, and 'high': $5 \times 10^4$ cells mL$^{-1}$) in a fully crossed design, creating a total of 12 treatments. There were seven replicates for each treatment. The control temperature (26 °C) represents the low average sea surface temperature that larval CoTS experience on the Great Barrier Reef (GBR) (https://apps.aims.gov.au/ts-explorer/) and ambient seawater was pH 8.0. The combined 30 °C, pH 7.6 treatment represents the upper levels for near-future warming and acidification on the GBR predicted by 2100[72]. The high-temperature treatment is already exceeded during warming anomalies that are increasing in frequency[73,74]. The low food level, which is equivalent to 0.1 μg chlorophyll a L$^{-1}$, reflects the lower end of natural

background levels of phytoplankton on the GBR (0.1–1.0 µg chlorophyll a L⁻¹)[29]. The high food level, which is equivalent to 5 µg chlorophyll a L⁻¹, reflects conditions that are considered eutrophic on the GBR[28,29]. The switch food level examines how larvae cope with a period of low rationing followed by high food availability, mimicking spatial or temporal patchiness in phytoplankton abundance which occurs in contemporary oceans (e.g., due to terrestrial nutrient inputs[29,31]), and which are likely to become more frequent due to climate change[2,9].

The experiment was conducted in a purpose-built flow-through seawater system[75]. In brief, experimental pH was manipulated using an automated CO₂ injection system in 60-L reservoirs where pH was lowered via CO₂ injection (pH 7.6 treatment) or not manipulated (control, pH 8.0 treatment). Air was continuously bubbled into reservoirs to aid mixing and maintain dissolved oxygen >98%. Experimental seawater was fed into subsequent 20-L reservoirs and warmed (26 or 30 °C) using aquarium heaters. Experimental seawater was delivered to replicates via individual dripper valves at a rate of 2.9 L h⁻¹ (~60 turnovers per day). To maintain even temperature and pH levels within each treatment, seawater was continuously recirculated from the reservoirs to the dripper valves and back using 8 W aquarium pumps. The pH and temperature were regulated so that experimental conditions were achieved in the replicate-rearing containers.

Larvae were reared in flow-through downwellers. Each downweller consisted of a PVC pipe (89 mm Ø, 235 mm high) that had a 74-µm mesh floor to retain the larvae. The pipe sat on short PVC stands inside a cylindrical polypropylene canister (125 mm Ø, 200 mm high). Experimental seawater was delivered to the PVC pipe via a dripper valve, where the water flowed down through the mesh floor and then spilled up over the top of the outer canister, thus maintaining a constant volume of water in the PVC section of the downweller (1.16 L). The inner PVC pipe was exchanged every 10 days or sooner if the mesh became partially obstructed.

Temperature, pH, and salinity were measured most days (4–6 times per week) in three haphazardly selected downwellers per treatment using a Hach® HQ40d multiprobe, calibrated with Tris buffers and high precision Hach conductivity standards. All pH measurements were done on the total scale (pH$_T$). Seawater samples for measuring A$_T$ (total alkalinity) were collected at the same time that pH was measured, filtered (0.45 µm) and kept at 4 °C in plastic storage vials following the protocol of Mos et al.[76] and used to determine total alkalinity (A$_T$) by potentiometric titration using a Metrohm 888 Titrando calibrated with a certified reference material (Batch 116)[77]. The partial pressure of dissolved CO₂ (pCO₂), bicarbonate (HCO₃⁻), carbonate (CO₃²⁻), and calcite saturation states (ΩCa) were calculated using salinity, temperature, pH$_T$, and A$_T$ values measured during the experiment using CO2SYS[78] and dissociation constants of Mehrbach et al.[79] as refitted by Dickson and Millero[80]. Mean experimental conditions are included in Supplementary Table 3.

**Study species.** Adult *Acanthaster* sp. were collected from the GBR near Cairns, Queensland and transported to Coffs Harbour, Australia. They were maintained in flow-through seawater at the temperature of their point of collection (25–27 °C). Due to the equivocal nomenclature of the Pacific species of *Acanthaster*[81,82], we refer to the species we investigated from the GBR as *Acanthaster* sp. or CoTS.

Gametes were collected from four females and four males using standard protocols[28–30,40,42]. Pieces of gonad were dissected from the aboral side of adult CoTS (~20–30 cm Ø) and placed in separate glass crystallising dishes. The male gonads immediately released sperm which was collected and stored dry at 4 °C before use. The female gonads were rinsed in 1-µm-filtered seawater (FSW) to remove immature eggs and then soaked in $10^{-5} M$ 1-methyl-adenine to induce ovulation. After ~1 h when large numbers of eggs were released from the gonads, the eggs were collected, rinsed in FSW, and checked microscopically for shape and integrity.

Gametes were fertilised in experimental water, which consisted of two temperatures (26 and 30 °C) factorially crossed with two pH levels (pH 8.0, 7.6), making four treatments in total. To fertilise the eggs, approximately equal numbers of eggs from each of the four females were mixed, divided into four groups, and added to 2-L glass beakers containing experimental seawater. Sperm from the four males was checked for motility, and then mixed in equal proportions in pH 8.0 or pH 7.6 seawater and counted using a haemocytometer. Sperm was added to eggs in the corresponding experimental seawater to create a ratio of approximately 100:1 (sperm:eggs). The eggs were checked for fertilisation using an Olympus BX53 microscope. Once >90% fertilisation had been achieved, the fertilised embryos were added to downwellers in their respective pH-temperature treatment at a density of 7.8 embryos mL⁻¹. When larvae were ready to begin feeding (~72 h post-fertilisation), the number of larvae in each downweller was adjusted to standardise density (1.46 ± 0.06 larvae mL⁻¹). Downwellers were then randomly assigned to food treatments.

**Microalgae and feeding.** Feeding began once the larvae had a complete digestive tract (~72 h post-fertilisation). Larvae were fed the tropical cryptomonad *Proteomonas sulcata* (CS-412, CSIRO culture collection, Tasmania) three times daily at a rate that corresponded to their assigned food treatment; low, high, or switch. *P. sulcata* was grown in seawater at 25–32 °C in aerated 20-L carboys, and fertilised with F media (AlgaBoost, AusAqua). *P. sulcata* was rapidly flushed from the downwellers (<20% remaining after 30 min), but CoTS larvae in all food treatments always had microalgae present in their gut 0–3 h after *P. sulcata* was fed.

**Data collection.** The density of larvae in each replicate was measured at the beginning of feeding (~72 h) and once a week thereafter. Each downweller was gently mixed to evenly distribute larvae, and five 5 mL subsamples were taken via pipette. The volume of the subsamples was increased over time as larval density declined due to mortality. Subsamples were viewed under an Olympus SZX7 dissecting microscope and the number of all living larvae counted. The mean density of the five subsamples was used as the value for each replicate in subsequent analyses. Survival at each time point was calculated for each replicate using density values and reported as a percentage of the original density present at the beginning of feeding.

At 11 and 18 dpf, ~20–50 larvae from each downweller were haphazardly collected with a pipette and placed in a 1.5 mL Eppendorf tube with 7% MgCl₂ for 15 min to relax the larvae. Afterwards, ~0.25 mL of 10% formaldehyde-FSW solution was added to each tube for preservation[40]. Within 7 days of collection, all larvae in each sample were scored as abnormal or normal. Abnormal larvae were identified by an irregular shape, small size, or arrested development (Fig. 3). All normal larvae (median 20, range 5–21) were then photographed using a digital camera mounted on the dissecting microscope (MIchrome 20). Larvae were positioned flat to the plane of focus. Length and width of normal larvae were measured from photographs using ImageJ image analysis software[83] following the methods of Kamya et al.[40]. The developmental stage of each normal larva was also recorded (Fig. 4). Replicates with fewer than nine staged larvae were excluded from formal statistical analyses of larval development.

Competency to settle was assessed in subsamples of ~20–100 larvae taken from replicates every 1–2 days from 14 dpf. Larvae were deemed to be competent if they had reached the late brachiolaria stage and had a prominent rudiment[29], which was often accompanied by 'settlement' behaviours including touching and attachment to surfaces, as well as partial resorption of larval feeding structures. The number of replicates in which larvae attained competence and the number of days elapsed (dpf) was recorded. All replicates were followed until >50% of surviving larvae had become competent or all larvae had died.

**Statistics and reproducibility.** Data on larval length, width, and length:width ratio at 11 dpf, proportion of abnormal larvae at 11 and 18 dpf, and the proportion of early brachiolaria at 11 dpf were analysed by three-way permutational analysis of variance in PRIMER v6 (PERMANOVA[84]). Three-way ANOVAs were conducted using temperature (26, 30 °C), pH (8.0, 7.6), and food level (low, switch, high) as fixed factors, and rearing container (i.e., downweller) as the level of replication ($n = 7$; median 20 larvae measured per downweller). Data on larval survival during 1–3 weeks were analysed using a repeated measures ANOVA design, with temperature, pH, and food level as fixed factors, and week as a random factor. Downwellers were used as the level of replication and included in the model to account for the non-independence of replicates measured over multiple weeks. Data on survival beyond week 3 were not formally analysed because replicates in some treatments had reached the endpoint (>50% competency) by week 4. Data for the proportion of late brachiolaria at 18 dpf was only available for the pH 8.0 treatments due to high mortality and a high proportion of abnormal larvae in pH 7.6 treatments by this time, resulting in low replication ($n < 3$). Therefore, data for the proportion of late brachiolaria at 18 dpf were analysed by two-way ANOVA with temperature and food level as fixed factors. Data for larval length, width, and length:width ratio at 18 dpf could not be analysed by an orthogonal three-way ANOVA due to missing treatments associated with high mortality and a high proportion of abnormal larvae in some treatments. Therefore, these data were initially analysed by one-way ANOVA with a combined treatment (temperature-pH-food level) as a single fixed factor. Planned contrasts were used to look for statistical differences among treatments driven by main effects and two-way interactions, using a Bonferroni correction to account for multiple comparisons. There was no evidence of interactive effects of temperature, pH, and food treatment on length, width, or length:width ratio at 18 dpf. As the planned contrasts gave the same outcomes as three-way ANOVA where the three-way interaction was excluded from the model, the three-way ANOVAs were presented.

Assumptions of normality and heterogeneity of variance were examined graphically using Q-Q residual plots, values for skewness and kurtosis, and Kolmogorov–Smirnov and Shapiro–Wilk tests, using IBM SPSS v27[85,86]. Data for the proportion of abnormal larvae at 11 and 18 dpf, and survival during weeks 1–3 were not normally distributed, and transformation had little effect. This was expected given the nature of the data. The reader should take this into account when considering the outcomes of these analyses. All other data met assumptions. Box plots were used to identify outliers. Three outliers were identified for length 11 dpf, width 11 dpf, length:width 11 dpf, and survival week 1–3. As the exclusion of outliers in ANOVA analyses had little effect on the outcome, the results of analyses with outliers included are presented. For ANOVA analyses, pair-wise comparisons of untransformed data were generated using Euclidean distance and 9999 permutations of the raw data. The default sum of squares was used (Type III for ANOVA and Type I for repeated measures ANOVA). Post-hoc pair-wise tests were performed when ANOVA results indicated that there were significant differences within treatments with three levels. Monte Carlo values were used if the number of permutations was low (<100). For reporting post-hoc tests, > and < indicate significantly greater than or less than at $p < 0.05$ respectively, while = indicates no significant difference, $p > 0.05$.

To assess the potential for larvae to complete larval development in the temperature-pH-food treatments, a binomial value was assigned to each downweller: either 0 = no larvae became competent to settle or 1 = at least one larva became competent. Data were analysed using binary logistic regression in IBM SPSS v27 software[86]. Competency was tested against the predictor variables 'temperature' (categorical), 'pH' (categorical) and 'food treatment' (categorical), and all interactions among these predictors. Model $\chi^2$ tests were used to select the best-fitting model out of models containing all possible combinations of these predictors and interactions. Wald $\chi^2$ tests were used to assess the significant effects of predictors ($p < 0.05$), and 95% bias-corrected and accelerated confidence intervals (BCa) were calculated based on 1000 bootstrap samples. Data on the number of days taken for larvae to reach competency was not formally analysed.

**Reporting summary**. Further information on research design is available in the Nature Portfolio Reporting Summary linked to this article.

## Data availability

The numerical source data for the charts/graphs are available free to download from the Dryad Digital Repository: https://doi.org/10.5061/dryad.msbcc2g2f[87]. All other data are available upon reasonable request from the corresponding author.

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

## Acknowledgements

The authors thank the Australian Marine Tourist Operators Association for their assistance in obtaining adult CoTS. This project was funded by an Australian Research Council (ARC) Discovery Indigenous grant to B.M. and S.A.D. (IN2000100026). B.M. is supported by an ARC-funded DAATSIA (Discovery Aboriginal and Torres Strait Islander Award, IN2000100026).

## Author contributions

B.M. and S.A.D. conceived the experiment. B.M. and N.M. conducted experiments. N.M. collated the data. B.M. analysed the data. All authors contributed to writing and reviewing the manuscript, and gave final approval for publication.

## Competing interests

The authors declare no competing interests.
