## [Peer Review File · Communications Biology]

Reviewers' comments:

Reviewer #1 (Remarks to the Author):

The paper written by B. Mos and colleagues experimentally investigates the effects of food availability on the responses of CoTS larvae to ocean acidification and warming scenarios.

This is a very interesting and timely study. It nicely complements the bulk of research available on multiple stressors effects on early life stages of marine invertebrates and is also key in the study of the biology and ecology of CoTS, given its ecological importance related to the health of tropical coral reefs around Australia.

I have very little to say about this study, it has been well designed, the writing is clear, and the results are well presented and discussed afterwards. It was very nice to read the manuscript and make sense of the study already after the first read. I want therefore to thank the authors for the work and effort put into it.

I only have some minor comments and questions:

- 1) General question about the methods: How fast do CoTS larvae react to changes in food concentrations – morphologically – compared to other echinoderms such as sea urchin larvae? And how was this taken into consideration when adjusting food rations in the switch treatment and subsequently measuring larval sizes?
- 2) Line 149 – did the survival calculations include only normal larvae, or also abnormal? This question relates to whether the increased survival of larvae at 30 degrees – line 232 – included abnormal larvae too. Please clarify.
- 3) Figure 1 - larval lengths and widths decrease with time, they seem to be generally lower at 18dpf than 11dpf, is this correct? Please check the scales. Given the information in figure 4, one would expect to find slightly bigger larvae at 18dpf than 11dpf, since there seems to be development towards the next stages with time.
- 4) Figure 3 – please explain how the survival data was calculated – given it was a cross factorial design, what do the data in the figure represent? How can we conclude on the effects of individual drivers (in this case temperature and acidification separately) and avoid confounding effects of the other drivers (such as food concentration) in the survival data? A repeated measures ANOVA was used for the survival data using pH, temperature and food concentration as fixed factors, yet the conclusions on figure 3 only include pH and temperature and there is no mention of food concentrations. Please revise/explain. Also, it would be clearer for the reader (who might not check the supplementary material) if the section on data analysis would be expanded in the main text of the manuscript to clarify this.
- 5) Line 344-347 – when measuring the lengths and widths of the larvae, the differences found in sizes, could they be due to different developmental stages? Or were they measured taking into account the same developmental stage within the same sample? Is it appropriate to talk about differences in sizes when comparing different stages? Please clarify in the text.

Thanks,
Nadjejda Espinel-Velasco
Institute for Arctic and Marine Biology
The Arctic University of Norway, Tromsø

Reviewer #2 (Remarks to the Author):

This research on the larvae of an ecologically important starfish (COTS) looks at the impacts of variable food availability in larvae exposed to acidification and increased temperature as well as in present day conditions. The study is interesting in being framed with respect to the predicted decrease in phytoplankton – the food of the larvae due to climate change. Although there are many studies that have investigated variable food regimes and food levels and marine larvae, I do not think any have highlighted these implications with respect to climate change. This is important. That said, the link to climate change in the habitat of the species was not well explained. The finding that increased food protects larvae over time at increased temperature is an important findings. I wonder if there area any other examples of this in marine invertebrates in general.

There are many studies of the impacts of food variation on marine larvae. The authors need to place their study with respect to previous studies on starfish larvae and larvae of other invertebrates.

The work builds on previous work by these and other authors on:

1. the impacts of climate change on COTS larvae and
2. food variability and the success of marine invertebrate larvae

The impacts of climate change on COTS larvae

The novel aspect is use of variable food as a factor combined with acidification and temperature increase both which were previously investigated with COTS larvae. It is not clear how this relates to what previous studies have found to understand the novel contribution of this study. I am a bit concerned about the indication on L. 310 that experimental variation (artefacts??) may confound comparisons. The abstract L 26 also mentions “may be more or less successful than indicated by previous ocean change studies...” and also highlighted in the conclusion L 388. “More or less” is not at all clear and needs to be explained. The authors need to add a table showing the findings of previous studies of COTS larvae, for instance what pH levels were used and what was the outcome, what temperatures were used and what were the outcomes, how long were the larvae reared for, what food levels were used, what species of phytoplankton were used etc. This is needed to show how the new findings in the present study compare. Please include a take home message on this to allow the reader to judge the results of the present and previous work. I am not sure where this table is best placed. The paragraph in the introduction L 66 needs assistance as does some of the comparisons in the discussion. It is possible that this table could be in the supplementary material.

Food variability and the success of marine invertebrate larvae

There are many papers on the impacts of variable feeding on marine larvae. The main findings here with respect to 1) low and high food ration and 2) recovery with food pulses have been documented in previous studies. Papers on impacts of changes in feed regimes with marine invertebrate larvae should be consulted to connect with the literature and make this work of broader interest. Some mention of this in the introduction is warranted to help set the scene for this work – beyond COTS.

At line 341 5 papers #58-62 are cited in this regard, but these include papers on frogs, spiders and mosquitoes. It would be far better to stick to papers on marine invertebrate larvae including papers on starfish larvae:

Allison 1994 Mar Biol 118; Basch 1996 Mar Biol 126, Pace et al 2008 JEMBE 353.

The authors used the same measurement as George (1999) did for *Pisaster*. Did the larvae change shape – have a plastic response as shown for *Pisaster* fed a range of diets (JEMBE 237).

There are papers on the larvae of other groups – mainly mollusks and crustaceans where larval diet was manipulated to assess outcomes for larval performance – here are just a few (there are many) that could be considered.

Howard and Hentschel 2005 L&O 50
Allen and Marshall 2010 Oikos
Chiu et al 2007 MEPS 343
Chiu et al 2008 Mar Biol 154
Emlet and Sadro 2006 ICB 46
Pechenik and Tyrell 2015 Mar Biol 162
Pechenik et al., 2002 JEMBE 280

Rational for the study link to climate change in the habitat of COTS larvae

It is well known that larvae encounter varying environmental conditions including food availability, patches of food at different depths (e.g. Metaxas papers on echinoderm larvae – Mar Biol 98) so it will be important for the authors to provide some ecological relevance to their experiments. For instance, the Feehan et al. 2018 Sci Rep paper on thermal stress as occurs in nature and food in a sea urchin larva. This is probably the most recent – climate related recent paper.

How do the food pulses in the experiments relate (or not) to conditions in the field. In consideration of nutrient runoff and excess food – will COTS larvae in the region experience food limitation. It seems pulses of food are routine for this species as mentioned on L 61.

The use of IPCC guidelines is indicative. What are the present temperature levels in the habitat of the species? The region is prone to heat waves (L. 376). What was the temperature of the water in recent heat waves during the spawning season of COTS and what would the authors suggest would have been the outcome for the larvae? This could be placed with respect to the “multiple scenarios” L. 388.

Other comments

The methods need to be clarified. It is not clear if the temperature, pH and food conditions are in a crossed design. The figures would indicate not, but some of the text indicates that they might have been. In the sentence on L. 92 – add the words “not crossed” – or something to this effect if this is the case. Time was also a factor as interactions between time and treatment are mentioned. State that time is also a factor. The experiments were run to day 18 – this is important to highlight in the comparisons with other studies especially as time in conditions is crucial – the legacy of time (e.g. Pechenik).

The temperature history of the parents before the gametes were collected is important to mention as this may influence outcomes.

Reviewer #3 (Remarks to the Author):

The manuscripts “Variable food availability alters the response of larval crown-of-thorns starfish to warming, but does not influence their response to reduced pH” aims at evaluating how food availability will impact the performance of Crown-of-Thorns seastar larvae towards ocean acidification and warming.

The authors provided a study analyzing survival, abnormalities, growth and morphology (allometry) of larvae in a fully crossed design of the three variables (food, temp, pH) tested. In general, I think it is a study that is of interest in the field of OA and ocean warming. Personally, I think that studies on the combination of several stressors are still very much in need, since they are difficult and challenging to perform. So, I think the study is relevant for the broader climate change community.

In general I had difficulties to follow the story in the manuscript partly because the results coming from interactive effects from three variables are complex, partly because effects are sometimes not

super clear and partly because the manuscript is sometimes very detailed, and therefore obscure the broader story.

I therefore suggest a few improvements on the order of the results, presented, and structure of the discussion. The most "difficult" part for me (usually working with larvae with up to 80 days larval duration) was to figure out, that CoTS larvae in control conditions would settle after 20 days and therefore any data beyond this time point in controls would be missing – in the beginning, I thought all the larvae had died at 26°C and therefore survival was better in high temperature. So, I suggest to have a short description of CoTS larval development in the introduction and/or state the aspect of survival very clearly in the results for readers that are not familiar with this model system.

Introduction:

The introduction is well written and structured. As suggested above, maybe a short description of larval development (duration) would be useful.

Line 79: In this paragraph, the three major hypothesis may be numbered, so that the text could follow these core ideas in the results and the discussion.

Material & Methods:

1) Line 128: could the authors detect any effect of temp/pH on fertilization, that could translate in the observed effects in the later development?

2) Line 138: I'm skeptical on the flow-through system the larvae were reared in because the food was (mostly) flushed away within 30 minutes after feeding. In my opinion, this does not resemble the way of how food would naturally be available in the field. However, the authors state later (line 310 f), that larvae from their experiments grew larger than in static treatments with same constant food conditions, so they seem to be healthy? From my own experience, we had a lot of difficulties in survival rates in flow-through systems for larvae leading to some replicates just dying while some had good survival rates. Survival rates were in my cases much better in static cultures. Apparently in this study, larvae of some replicates were also dying (Figure 5 a) and I am wondering, if survival in the remaining replicates were high or low and how survival rates were calculated. Maybe the authors could comment on this aspect, and if larval survival was adequate for the experiments with CoTS larvae? More specifically, that the high mortality in some replicates is not primarily due to experimental set up rather than real treatment effects?

3) Line 156: if morphology was only done on "normal" larvae, then the morphology data would already be biased due to high abnormality rates in some treatments. The authors should include a statement for this in the manuscript, so that the reader is aware of this experimental "problem".

4) Statistics: were all the requirements for a parametric ANOVA met (equal variance, homogeneity?). Was a repeated measure ANOVA or three-way ANOVA conducted? Please specify statistical methods a bit more in detail.

Results:

a) Initially, I got confused, what the story of the paper was about, because results were discussed in great detail and are very complex due to the nature of the fully crossed three variable design. So, I think, a shortening and focusing of results on the main ideas of the manuscript (Hypothesis from Introduction) would be helpful for the reader to follow. From the introduction I got the impression that the focus of the hypotheses was laid on survival and overall development. I, therefore, suggest the authors to change the order of results and/or combine results to:

1. survival: here I find the data from the supplementary material much better than the "global – no food related" data of the current figure 3, because the story of the paper is on food availability and the survival data should also include the food treatments. The authors could think about calculating a mortality rate (dead larvae per week?). The text should state clearly, that settlement is reached after 20 days in control conditions. As mentioned above, it took me some time to understand, that the larvae at 26°C, pH8 were not all dead after 3 weeks, but were already competent for settlement and therefore the graphs ended here. So, I can imagine it would be helpful for the reader to present

survival and settlement data together and to clearly state, why data at 26°C end after 3 weeks.

2. Proportion of abnormal development could be combined with the larval staging and morphology to have a full presentative figure of "larval development".

3. Larval morphology: a correlation between larval length and width would help to identify, if larvae were altogether only smaller in some treatments or if the allometry of the larvae was changing in response to treatments. It would also help to focus the manuscript onto the broader perspective given in the hypothesis of the introduction. The results for this part is extremely long and detailed compared to the other result figures. As mentioned above (Mat/Methods point 3), the text should make clear, that in some treatments (30°C, pH 7.6 low food) a high rate of abnormal larvae (60%) would lead to some bias (as for 26°C, pH 7.6. where no data was presented because of high abnormality).

b) Finally, to me there is an inconsistency in the results between Figure 1 and Figure 5: In figure 1, there is no morphology data available for high food, 26°C, pH 7.6 (18dpf), in Figure 5, in contrast, there is data available for this treatment for competent larvae. In Figure 5, data is missing for low food the same treatment, while there is data in figure 1 for this treatment and timepoint. I'm wondering, how larvae can become competent in the high food treatment, but there is no data for larval morphology? Or if there has been a mistake in the data?

Discussion:

The first long paragraph (line 282 to 304) reads to me like a repetition of the results – maybe use this for results of main hypotheses?

Thereafter, the discussion is good to follow, but could benefit from a repetition of the three hypothesis (from introduction) as introductory sentence into each of the paragraphs. It would help the reader to focus on the core ideas after the rather complex and diverse results.

Since the main objective of the manuscript, as I understood, is to elucidate how feeding impacts larval fitness in response to future ocean change, I think, the authors should discuss the matter of feeding a little bit more detailed in the discussion.

At line 374, the authors mention that the mechanisms how low seawater pH may limit feeding rates are not known. This is not quite right. In fact, there are publications (some are mentioned in the cited review #68) on sea urchin larvae demonstrating that low seawater pH causes a reduction in gastric pH which in turn is causing a decrease in digestive efficiency, because digestive enzyme pH optima are at high gastric pH (Stumpp et al 2013, full reference below). That OA is causing a decrease in gastric pH is also true for sea star *Asterias rubens* larvae (Hu et al. 2018) and may potentially also be true for CoTS and could lead to some of the feeding effects presented in the study, although the link between digestive enzymes and low midgut pH was not tested in sea star larvae. Finally, the capacity of gastric pH regulation in echinoderm larvae (including sea stars) correlates with sensitivity to OA (Hu et al 2017, reference also below). The literature may be relevant, if the authors discuss OA impacts on energetics/feeding in echinoderm larvae and would like to discuss the mechanisms of OA impacts on feeding physiology in more detail. So this may indeed be a factor, why the mortality rate was so high in CoTS larvae held at 26°C and pH 7.6, and that feeding regimes could rescue some of the 30°C larvae kept at 7.6°C from the low to switched food treatment, when considering that enzymatic activity also increases along with temperature – it is just a hypothesis, but it is intriguing to see that at 30°C and low pH treatment the proportion of abnormal larvae decreases and survival is increased when exposed to higher food treatments, although settlement is not improved.

In general, I think, that the interactive effects of temperature, OA and food is very complex and it is challenging to write a clear story. Nevertheless, studies with several stressors are important in this research field, because the majority of studies address only the impact of one or two stressors on larval development.

I hope, the editors and authors find my suggestions for improvement of the manuscript useful.

References:

Hu, M.Y.; Lein, E.; Bleich, M.; Melzner, F.; Stumpp, M. (2018) Trans-life cycle acclimation to experimental ocean acidification affects gastric pH homeostasis and larval recruitment in the sea star *Asterias rubens*. *Acta Physiologica* doi: 10.1111/apha.13075

Hu, M.Y.; Tseng, Y.C.; Su, Y.H.; Lein, E.; Lee, J.R.; Dupont, S.; Stumpp, M. (2017) Variability in larval gut pH regulation defines sensitivity to ocean acidification in six species of the ambulacraria superphylum. *Proceedings of the Royal Society London B (Biol)* doi: 10.1098/rspb.2017.1066

Stumpp M.; Hu. M.Y.; Casties I.; Saborowski R.; Bleich M.; Melzner F.; Dupont S. (2013) Digestion in sea urchin larvae impaired under ocean acidification. *Nature Climate Change*, 3:1044-1049

Response to referees:

We thank the reviewers for their kind words and suggestions that have improved the manuscript. We have made all of the changes that were suggested except where these conflict with the formatting requirements or policies of *Communications Biology*.

Note: line numbers refer to the ‘revised with tracked changes’ version of the manuscript.

Reviewer 1 (Nadjeđa Espinel-Velasco)	Response to Referee’s Comments
The paper written by B. Mos and colleagues experimentally investigates the effects of food availability on the responses of CoTS larvae to ocean acidification and warming scenarios. This a very interesting and timely study. It nicely complements the bulk of research available on multiple stressors effects on early life stages or marine invertebrates and is also key in the study of the biology and ecology of CoTS, given its ecological importance related to the health of tropical coral reefs around Australia. I have very little to say about this study, it has been well designed, the writing is clear, and the results are well presented and discussed afterwards. It was very nice to read the manuscript and make sense of the study already after the first read. I want therefore to thank the authors for the work and effort put into it. I only have some minor comments and questions:	NO CHANGES REQUESTED. We thank the reviewer for their input.
1) General question about the methods: How fast do CoTS larvae react to changes in food concentrations – morphologically – compared to other echinoderms such as sea urchin larvae? And how was this taken into consideration when adjusting food rations in the switch treatment and subsequently measuring larval sizes?	DONE. Our previous experiences raising CoTS suggested larvae respond rapidly to changes in food levels - within days (also see publications such as those led by Wolfe and Kamyra on the responses of CoTS larvae to food, temperature and/or acidification over time). CoTS has a relatively quick larval phase for an echinoderm (13-17 days under optimal conditions) but can survive for many weeks without any food. We selected 7 days low food and 7 days high food as the switch treatment to allow us to maximise the duration of low food condition (and thereby see a measurable negative effect) but still allow us to capture and compare morphology of larvae in all treatments before larvae in high food treatments began to settle. We now provide more context about CoTS larval development in lines 76-79.
2) Line 149 – did the survival calculations include only normal larvae, or also abnormal? This question relates to whether the increased survival of larvae at	DONE. Survival was calculated including both abnormal and normal larvae. We have clarified this in the Methods (line

30 degrees – line 232 – included abnormal larvae too. Please clarify.	502-509) and figure captions (Figure 1, line 857) and in the Supplementary Material.
3) Figure 1 - larval lengths and widths decrease with time, they seem to be generally lower at 18dpf than 11dpf, is this correct? Please check the scales. Given the information in figure 4, one would expect to find slightly bigger larvae at 18dpf than 11dpf, since there seems to be development towards the next stages with time.	DONE. We re-examined the figures and the underlying data to ensure they are correct. We confirm the figures are correct as they appear. The apparent decrease in larval size from 11–18 dpf in some treatments may be due to the effects of the low food ration (starvation) and the tendency for smaller and abnormal larvae to survive longer than large, well-developed larvae in stressful environments.
4) Figure 3 – please explain how the survival data was calculated – given it was a cross factorial design, what do the data in the figure represent? How can we conclude on the effects of individual drivers (in this case temperature and acidification separately) and avoid confounding effects of the other drivers (such as food concentration) in the survival data? A repeated measures ANOVA was used for the survival data using pH, temperature and food concentration as fixed factors, yet the conclusions on figure 3 only include pH and temperature and there is no mention of food concentrations. Please revise/explain. Also, it would be clearer for the reader (who might not check the supplementary material) if the section on data analysis would be expanded in the main text of the manuscript to clarify this.	DONE. We agree our figure caption did a poor job of explaining the survival data presented in Fig. 3. We have updated the figure caption to better explain the outcomes of the statistical analysis and data presented. (Note Figure 3 is now Figure 1, also see responses below). We have moved the requested information on statistical analysis from Supplementary Material into the Methods section (lines 529-575).
5) Line 344-347 – when measuring the lengths and widths of the larvae, the differences found in sizes, could they be due to different developmental stages? Or were they measured taking into account the same developmental stage within the same sample? Is it appropriate to talk about differences in sizes when comparing different stages? Please clarify in the text.	DONE. The text has been edited to better explain how larvae were measured. Size and development in CoTS are not always directly linked because larvae are not calcified and are highly plastic.
Thanks, Nadjeđa Espinel-Velasco Institute for Arctic and Marine Biology The Arctic University of Norway, Tromsø	
Reviewer 2	
This research on the larvae of an ecologically important starfish (COTS) looks at the impacts of variable food availability in larvae exposed to acidification and increased temperature as well as in present day conditions. The study is interesting in being framed with respect to the predicted decrease in phytoplankton – the food of the larvae due to climate change. Although there are many studies that have investigated variable food regimes and food levels and marine larvae, I do not think any have highlighted these implications with respect to	NO CHANGES REQUESTED. We thank the reviewer for their input.

climate change. This is important. That said, the link to climate change in the habitat of the species was not well explained. The finding that increased food protects larvae over time at increased temperature is an important findings. I wonder if there area any other examples of this in marine invertebrates in general.	
There are many studies of the impacts of food variation on marine larvae. The authors need to place their study with respect to previous studies on starfish larvae and larvae of other invertebrates. The work builds on previous work by these and other authors on:  1. the impacts of climate change on COTS larvae and 2. food variability and the success of marine invertebrate larvae 	NO CHANGES REQUESTED.
The impacts of climate change on COTS larvae The novel aspect is use of variable food as a factor combined with acidification and temperature increase both which were previously investigated with COTS larvae. It is not clear how this relates to what previous studies have found to understand the novel contribution of this study. I am a bit concerned about the indication on L. 310 that experimental variation (artefacts??) may confound comparisons. The abstract L 26 also mentions “may be more or less successful than indicated by previous ocean change studies...” and also highlighted in the conclusion L 388. “More or less” is not at all clear and needs to be explained. The authors need to add a table showing the findings of previous studies of COTS larvae, for instance what pH levels were used and what was the outcome, what temperatures were used and what were the outcomes, how long were the larvae reared for, what food levels were used, what species of phytoplankton were used etc. This is needed to show how the new findings in the present study compare. Please include a take home message on this to allow the reader to judge the results of the present and previous work. I am not sure where this table is best placed. The paragraph in the introduction L 66 needs assistance as does some of the comparisons in the discussion. It is possible that this table could be in the supplementary material.	DONE. We have removed the phrase ‘more or less’ from the Abstract and the Discussion, and rewritten the sections referred to by the reviewer to be clearer about the novel contribution of this study. DONE. This is an excellent idea, however Hue et al., 2022 has recently published a table on OA/OW effects on CoTS and Wolfe et al., 2017 has published a table on phytoplankton experiments with CoTS. We cite these reviews to enable the reader to more easily find and refer to these tables.

Food variability and the success of marine invertebrate larvae There are many papers on the impacts of variable feeding on marine larvae. The main findings here with respect to 1) low and high food ration and 2) recovery with food pulses have been documented in previous studies. Papers on impacts of changes in feed regimes with marine invertebrate larvae should be consulted to connect with the literature and make this work of broader interest. Some mention of this in the introduction is warranted to help set the scene for this work – beyond CoTS. At line 341 5 papers #58-62 are cited in this regard, but these include papers on frogs, spiders and mosquitoes. It would be far better to stick to papers on marine invertebrate larvae including papers on starfish larvae: Allison 1994 Mar Biol 118; Basch 1996 Mar Biol 126, Pace et al 2008 JEMBE 353. The authors used the same measurement as George (1999) did for Pisaster. Did the larvae change shape – have a plastic response as shown for Pisaster fed a range of diets (JEMBE 237). There are papers on the larvae of other groups – mainly mollusks and crustaceans where larval diet was manipulated to assess outcomes for larval performance – here are just a few (there are many) that could be considered. Howard and Hentschel 2005 L&O 50 Allen and Marshall 2010 Oikos Chiu et al 2007 MEPS 343 Chiu et al 2008 Mar Biol 154 Emlet and Sadro 2006 ICB 46 Pechenik and Tyrell 2015 Mar Biol 162 Pechenik et al., 2002 JEMBE 280	ADDRESSED. We thank the reviewer for the suggested citations to add but we cannot fit more citations in the manuscript without substantially exceeding Communications Biology word limit. Given the broad readership of Communications Biology, we feel it is appropriate to include citations on frogs, spiders and mosquitoes to ensure our work is of broader interest beyond CoTS. We welcome further guidance from the editor should additional citations be needed or appropriate. We did not specifically examine whether the larvae changed shape but they may have given CoTS larvae are plastic in response to variation in food.
Rational for the study link to climate change in the habitat of CoTS larvae It is well known that larvae encounter varying environmental conditions including food availability, patches of food at different depths (e.g. Metaxas papers on echinoderm larvae – Mar Biol 98) so it will be important for the authors to provide some ecological relevance to their experiments. For instance, the Feehan et al. 2018 Sci Rep paper on thermal stress as occurs in nature and food in a sea urchin larva. This is probably the most recent – climate related recent paper.	DONE. We have added text and references to provide further context about food availability for CoTS in the natural environment (lines 452-464). We note understanding of food availability at the scale experienced by larvae is poor and acknowledge laboratory experiments such as ours are only ever a poor approximation of nature.

How do the food pulses in the experiments relate (or not) to conditions in the field. In consideration of nutrient runoff and excess food – will COTS larvae in the region experience food limitation. It seems pulses of food are routine for this species as mentioned on L 61	ADDRESSED. We believe it would be inappropriate to speculate on how nutrient runoff and excess food might affect CoTS larvae as this was outside the scope of this study. Rather, given that this paper shows important ways in which variable food can impact larvae, others may want to do similar work specifically tailored to the variability that is documented in a particular habitat.
The use of IPCC guidelines is indicative. What are the present temperature levels in the habitat of the species? The region is prone to heat waves (L. 376). What was the temperature of the water in recent heat waves during the spawning season of COTS and what would the authors suggest would have been the outcome for the larvae? This could be placed with respect to the “multiple scenarios” L. 388.	DONE. We have provided additional information about the temperatures experienced by the broodstock in Supplementary Methods and refer to this in the main text. We believe it would be inappropriate to speculate about the fate of larvae exposed to recent heat waves as this is outside the scope of our study (e.g. stable temperatures in our experiments vs. variable heat wave conditions in nature), but is something that could be examined in future studies.
Other comments The methods need to be clarified. It is not clear if the temperature, pH and food conditions are in a crossed design. The figures would indicate not, but some of the text indicates that they might have been. In the sentence on L. 92 – add the words “not crossed” – or something to this effect if this is the case. Time was also a factor as interactions between time and treatment are mentioned. State that time is also a factor. The experiments were run to day 18 – this is important to highlight in the comparisons with other studies especially as time in conditions is crucial – the legacy of time (e.g. Pechenik).	DONE. We have edited the text to better explain our experimental design (e.g. Line 449-464).
The temperature history of the parents before the gametes were collected is important to mention as this may influence outcomes.	DONE. This information has been added (also see previous comment).
Reviewer 3:	

The manuscripts “Variable food availability alters the response of larval crown-of-thorns starfish to warming, but does not influence their response to reduced pH” aims at evaluating how food availability will impact the performance of Crown-of-Thorns seastar larvae towards ocean acidification and warming. The authors provided a study analyzing survival, abnormalities, growth and morphology (allometry) of larvae in a fully crossed design of the three variables (food, temp, pH) tested. In general, I think it is a study that is of interest in the field of OA and ocean warming. Personally, I think that studies on the combination of several stressors are still very much in need, since they are difficult and challenging to perform. So, I think the study is relevant for the broader climate change community.	NO CHANGES REQUESTED. We thank the reviewer for their input.
In general I had difficulties to follow the story in the manuscript partly because the results coming from interactive effects from three variables are complex, partly because effects are sometimes not super clear and partly because the manuscript is sometimes very detailed, and therefore obscure the broader story. I therefore suggest a few improvements on the order of the results, presented, and structure of the discussion. The most “difficult” part for me (usually working with larvae with up to 80 days larval duration) was to figure out, that CoTS larvae in control conditions would settle after 20 days and therefore any data beyond this time point in controls would be missing – in the beginning, I thought all the larvae had died at 26°C and therefore survival was better in high temperature. So, I suggest to have a short description of CoTS larval development in the introduction and/or state the aspect of survival very clearly in the results for readers that are not familiar with this model system. Introduction: The introduction is well written and structured. As suggested above, maybe a short description of larval development (duration) would be useful.	DONE. A brief description of the larval development of CoTS has been added to the Introduction (line 76-79).
Line 79: In this paragraph, the three major hypothesis may be numbered, so that the text could follow these core ideas in the results and the discussion.	TEXT REMOVED. To comply with Communications Biology Instructions to Authors, we rewrote the final paragraph of the Introduction which necessitated removing the hypotheses.

Material & Methods: 1) Line 128: could the authors detect any effect of temp/pH on fertilization, that could translate in the observed effects in the later development?	ADDRESSED. We did not measure the effects of pH and temperature on CoTS fertilisation. These factors do not affect fertilisation at the levels tested in this study (see Introduction and references cited there, lines 81-93).
2) Line 138: I'm skeptical on the flow-through system the larvae were reared in because the food was (mostly) flushed away within 30 minutes after feeding. In my opinion, this does not resemble the way of how food would naturally be available in the field. However, the authors state later (line 310 f), that larvae from their experiments grew larger than in static treatments with same constant food conditions, so they seem to be healthy? From my own experience, we had a lot of difficulties in survival rates in flow-through systems for larvae leading to some replicates just dying while some had good survival rates. Survival rates were in my cases much better in static cultures. Apparently in this study, larvae of some replicates were also dying (Figure 5) and I am wondering, if survival in the remaining replicates were high or low and how survival rates were calculated. Maybe the authors could comment on this aspect, and if larval survival was adequate for the experiments with CoTS larvae? More specifically, that the high mortality in some replicates is not primarily due to experimental set up rather than real treatment effects?	DONE. The methods used to calculate survival have been clarified (also see previous comments for Reviewer 1). Survival rates in this study were similar to previous OA/OW studies on CoTS and declined slowly over time as might be expected for larvae exposed to constant stress (low food, high temperatures, low pH).
3) Line 156: if morphology was only done on "normal" larvae, then the morphology data would already be biased due to high abnormality rates in some treatments. The authors should include a statement for this in the manuscript, so that the reader is aware of this experimental "problem".	DONE. We have added text to better emphasise morphology measurements were only done on normal larvae.
4) Statistics: were all the requirements for a parametric ANOVA met (equal variance, homogeneity?). Was a repeated measure ANOVA or three-way ANOVA conducted? Please specify statistical methods a bit more in detail.	DONE. We have moved relevant text from the supplementary material into the main manuscript (Also see previous comment for Reviewer 1).
Results: a) Initially, I got confused, what the story of the paper was about, because results were discussed in great detail and are very complex due to the nature of the fully crossed three variable design. So, I think, a shortening and focusing of results on the main ideas of the manuscript (Hypothesis from Introduction) would be helpful for the reader to follow. From the introduction I got the impression that the focus of the hypotheses was laid on survival and overall development. I, therefore, suggest the authors to change the order of results and/or	DONE.  1. We have edited the text to enhance clarity and have ordered the results following the suggestion of the Reviewer (also see previous responses to Reviewer 1). 2. We attempted to combine the abnormal and normal figures, but the resulting figure was too large and complex to comply with the figure requirements of Communications Biology.

combine results to:  1. survival: here I find the data from the supplementary material much better than the “global – no food related” data of the current figure 3, because the story of the paper is on food availability and the survival data should also include the food treatments. The authors could think about calculating a mortality rate (dead larvae per week?). The text should state clearly, that settlement is reached after 20 days in control conditions. As mentioned above, it took me some time to understand, that the larvae at 26°C, pH8 were not all dead after 3 weeks, but were already competent for settlement and therefore the graphs ended here. So, I can imagine it would be helpful for the reader to present survival and settlement data together and to clearly state, why data at 26°C end after 3 weeks. 2. Proportion of abnormal development could be combined with the larval staging and morphology to have a full presentative figure of “larval development”. 3. Larval morphology: a correlation between larval length and width would help to identify, if larvae were altogether only smaller in some treatments or if the allometry of the larvae was changing in response to treatments. It would also help to focus the manuscript onto the broader perspective given in the hypothesis of the introduction. The results for this part is extremely long and detailed compared to the other result figures. As mentioned above (Mat/Methods point 3), the text should make clear, that in some treatments (30°C, pH 7.6 low food) a high rate of abnormal larvae (60%) would lead to some bias (as for 26°C, pH 7.6. where no data was presented because of high abnormality). 	 3. Additional text has been added to better explain that data were not presented for some treatments due to low replication associated with high abnormality and/or mortality.
b) Finally, to me there is an inconsistency in the results between Figure 1 and Figure 5: In figure 1, there is no morphology data available for high food, 26°C, pH 7.6 (18dpf), in Figure 5, in contrast, there is data available for this treatment for competent larvae. In Figure 5, data is missing for low food the same treatment, while there is data in figure 1 for this treatment and timepoint. I’m wondering, how larvae can become competent in the high food treatment, but there is no data for larval morphology? Or if there has been a mistake in the data?	DONE. We have re-examined the graphs and underlying data and confirm they are correct. We believe the appearance of an inconsistency is due our clumsy explanation in the figure caption. We have edited the text to clarify why this data was not presented (also see previous responses).

The first long paragraph (line 282 to 304) reads to me like a repetition of the results – maybe use this for results of main hypotheses? Thereafter, the discussion is good to follow, but could benefit from a repetition of the three hypothesis (from introduction) as introductory sentence into each of the paragraphs. It would help the reader to focus on the core ideas after the rather complex and diverse results.	DONE. This section of text has been removed. To comply with Communications Biology Instructions to Authors, we rewrote the final paragraph of the Introduction which necessitated removing the hypotheses as previously written.
Since the main objective of the manuscript, as I understood, is to elucidate how feeding impacts larval fitness in response to future ocean change, I think, the authors should discuss the matter of feeding a little bit more detailed in the discussion. At line 374, the authors mention that the mechanisms how low seawater pH may limit feeding rates are not known. This is not quite right. In fact, there are publications (some are mentioned in the cited review #68) on sea urchin larvae demonstrating that low seawater pH causes a reduction in gastric pH which in turn is causing a decrease in digestive efficiency, because digestive enzyme pH optima are at high gastric pH (Stumpp et al 2013, full reference below). That OA is causing a decrease in gastric pH is also true for sea star Asterias rubens larvae (Hu et al. 2018) and may potentially also be true for CoTS and could lead to some of the feeding effects presented in the study, although the link between digestive enzymes and low midgut pH was not tested in sea star larvae. Finally, the capacity of gastric pH regulation in echinoderm larvae (including sea stars) correlates with sensitivity to OA (Hu et al 2017, reference also below). The literature may be relevant, if the authors discuss OA impacts on energetics/feeding in echinoderm larvae and would like to discuss the mechanisms of OA impacts on feeding physiology in more detail. So this may indeed be a factor, why the mortality rate was so high in CoTS larvae held at 26°C and pH 7.6, and that feeding regimes could rescue some of the 30°C larvae kept at 7.6°C from the low to switched food treatment, when considering that enzymatic activity also increases along with temperature – it is just a hypothesis, but it is intriguing to see that at 30°C and low pH treatment the proportion of abnormal larvae decreases and survival is increased when exposed to higher food treatments, although settlement is not improved.	DONE. We have changed ‘feeding’ to ‘ingestion’ to more accurately describe the mechanism. We have added text and now cite studies examining effects of acidification on larval digestion.

In general, I think, that the interactive effects of temperature, OA and food is very complex and it is challenging to write a clear story. Nevertheless, studies with several stressors are important in this research field, because the majority of studies address only the impact of one or two stressors on larval development. I hope, the editors and authors find my suggestions for improvement of the manuscript useful. References: Hu, M.Y.; Lein, E.; Bleich, M.; Melzner, F.; Stumpp, M. (2018) Trans-life cycle acclimation to experimental ocean acidification affects gastric pH homeostasis and larval recruitment in the sea star Asterias rubens. Acta Physiologica doi: 10.1111/apha.13075 Hu, M.Y.; Tseng, Y.C.; Su, Y.H.; Lein, E.; Lee, J.R.; Dupont, S.; Stumpp, M. (2017) Variability in larval gut pH regulation defines sensitivity to ocean acidification in six species of the ambulacraria superphylum. Proceedings of the Royal Society London B (Biol) doi: 10.1098/rspb.2017.1066 Stumpp M.; Hu. M.Y.; Casties I.; Saborowski R.; Bleich M.; Melzner F.; Dupont S. (2013) Digestion in sea urchin larvae impaired under ocean acidification. Nature Climate Change, 3:1044-1049	NO CHANGES REQUESTED.
OTHER CHANGES	
The first and corresponding author, Benjamin Mos, has moved institutions since the manuscript was first submitted. His affiliation and contact details have been updated.	
We have formatted the manuscript to ensure it complies with the formatting requirements, recommended word count, and policies of Communications Biology.	

REVIEWERS' COMMENTS:

Reviewer #1 (Remarks to the Author):

I thank the authors for their work in revising and clarifying the comments and questions the reviewers had related to the first version of the manuscript. I feel that the changes have positively improved the manuscript and I have no further comments or questions regarding this work.

Many thanks,

Nadjejda Espinel-Velasco
Institute for Arctic and Marine Biology
The Arctic University of Norway, Tromsø

Reviewer #2 (Remarks to the Author):

The authors have done a good job addressing my comments – although some suggestions seem to have been missed or not sufficiently considered. The clarifications in response to reviewer 1 and 3 on abnormal larvae prompts queries. This is interesting work especially as the larvae were taken to settlement.

In my first comment I asked if there were any other examples that show that increased food protects larvae over time and suggested other invertebrate references to follow up. I understand the word limits. If there is no room to accommodate more papers perhaps 1-2 of these papers could be substituted for 1-2 of the mosquitoes, frogs or spider papers cited. While I agree that Communications Biology readers appreciate broad citations – the authors do need to recognise that this type of work has been done before with other important starfish (*Pisaster*) by George. This is relevant to a comment of rev #3.

Figure 2 (length/width) indicates that there is a shape change as in other studies of larval morphological phenotypic plasticity, measured by the body length relative to the body width. This is important with respect to allometry (comment Rev #3). I recommend as suggested by Rev #3 that the length-width ratio is also plotted to address this possibility.

I appreciate mention of Hue et al 2022 – I did not know this work. The Hue paper is relevant for this manuscript and should be cited. The take home in Hue - adult environment-acclimation influences the response of COTS larvae to temperature and acidification is important to mention, could add at L 247.

The review table in Hue is helpful. There are not many comparative papers on larvae. The previous publications by the authors (eg Kamyra et al. 2014) are the most comparative where under similar conditions (food concentration, 30 dg) mortality was high and 100% abnormal larvae on day 10 – similar results.

L. 88 Need to add a sentence on acidification results to balance with sentences on temperature L 83-87.

L. 97 Does not match figure 1 where survival is above 14%.

The tendency for smaller abnormal larvae to survive longer and at L 156 79% abnormal in day 18 – prompts the question was this due to mass cloning? With what we know about echinoderm larval biology that seems very possible and should be acknowledged. A paper on starfish larval cloning (# 46) that shows food levels influence the incidence of cloning is cited but cloning is not mentioned, please expand.

L. 185 188 add x-x settled larvae per replicate tube – or in the figure legends give the range in the number of larvae in the tubes. Figure 5 is very interesting.

Reviewer #3 (Remarks to the Author):

Dear authors and editor,

I find that my comments have been met adequately. I have no further comments.

Response to referees:

We thank the editor and reviewers for their kind words and suggestions that have improved the manuscript.

Note: line numbers refer to the 'revised, without tracked changes' version of the manuscript.

Editor's comments	Response to Referee's Comments
Your manuscript entitled "Variable food alters responses of larval crown-of-thorns starfish to ocean warming, but not acidification" has now been seen again by our referees, whose comments appear below. In light of their advice I am delighted to say that we are happy, in principle, to publish a suitably revised version in Communications Biology under the open access CC BY license (Creative Commons Attribution v4.0 International License). We therefore invite you to revise your paper one last time to address the remaining concerns of our reviewers. Please address all the remaining concerns of Reviewer #2 in your revised manuscript. Note that your manuscript is currently well under our suggested word limit of 5,000 words for the main text (includes only the Introduction, Results, and Discussion). We also ask that you add your hypotheses back into the re-written final paragraph of the Introduction section, as it was not necessary to remove these from this paragraph.	We thank the editor and reviewers for their helpful suggestions and positive feedback. We have made all changes suggested by reviewer 2. We thank the editor for clarifying the word count. We incorrectly believed the word count included the Methods section. We have rewritten the hypotheses back into the Introduction.
At the same time we ask that you edit your manuscript to comply with our format requirements and to maximise the accessibility and therefore the impact of your work. * Please see the attached document for editorial requests for the final version (.docx file). Please ensure a completed version of this file is uploaded as a Related Manuscript with your final submission. * Please review our final submission file checklist to ensure all necessary files are present with your final submission and to avoid delays in accepting your manuscript. For your reference, a style and formatting guide is available here and includes all of our style requirements. * An updated editorial policy checklist that verifies compliance with all required editorial policies must be completed and uploaded with the revised	DONE. Completed version uploaded. Checklist completed. Manuscript formatted to match formatting guide. Editorial policy checklist completed.

manuscript. All points on the policy checklist must be addressed; if needed, please revise your manuscript in response to these points. Please note that this form is a dynamic 'smart pdf' and must therefore be downloaded and completed in Adobe Reader. https://www.nature.com/documents/nr-editorial-policy-checklist.pdf It is important that you pay careful attention to the requests in these documents to avoid a delay in formal acceptance of the article. If you have any questions or concerns about any of our requests, please do not hesitate to contact me.	
Reviewer 1: Nadjeida Espinel-Velasco	Author response to reviewer:
I thank the authors for their work in revising and clarifying the comments and questions the reviewers had related to the first version of the manuscript. I feel that the changes have positively improved the manuscript and I have no further comments or questions regarding this work.	We thank Reviewer 1 for their kind words and helpful suggestions.
Reviewer 2:	Author response to reviewer:
The authors have done a good job addressing my comments – although some suggestions seem to have been missed or not sufficiently considered. The clarifications in response to reviewer 1 and 3 on abnormal larvae prompts queries. This is interesting work especially as the larvae were taken to settlement. In my first comment I asked if there were any other examples that show that increased food protects larvae over time and suggested other invertebrate references to follow up. I understand the word limits. If there is no room to accommodate more papers perhaps 1-2 of these papers could be substituted for 1-2 of the mosquitoes, frogs or spider papers cited. While I agree that Communications Biology readers appreciate broad citations – the authors do need to recognise that this type of work has been done before with other important starfish (Pisaster) by George. This is relevant to a comment of rev #3.	We thank the reviewer for their comments and helpful feedback. We incorrectly believed the word count included the Methods section. We are sorry that this made it appear as if we had missed or ignored the reviewer's suggestions. This was not our intention. We added three citations suggested by the reviewer, including George 1999.
Figure 2 (length/width) indicates that there is a shape change as in other studies of larval morphological phenotypic plasticity, measured by the body length relative to the body width. This is important with respect to allometry (comment Rev #3). I recommend as suggested by Rev #3 that the length-width ratio is also plotted to address this possibility.	DONE: We plotted the length-width ratio at day 11 and day 18. There were no interactive effects of temperature, pH and food, but we present the graphs in the Supplementary Material for the interested reader.

I appreciate mention of Hue et al 2022 – I did not know this work. The Hue paper is relevant for this manuscript and should be cited. The take home in Hue - adult environment-acclimation influences the response of COTS larvae to temperature and acidification is important to mention, could add at L 247. The review table in Hue is helpful. There are not many comparative papers on larvae. The previous publications by the authors (eg Kanya et al. 2014) are the most comparative where under similar conditions (food concentration, 30 dg) mortality was high and 100% abnormal larvae on day 10 – similar results.	DONE. We meant to cite two papers from Hue et al., but mistakenly included only one in the reference list. This has been fixed. Hue et al. 2022 is now cited multiple times. We thank the reviewer for bringing this oversight to our attention.
L. 88 Need to add a sentence on acidification results to balance with sentences on temperature L 83-87.	DONE. A sentence on acidification has been added, Ln. 106.
L. 97 Does not match figure 1 where survival is above 14%.	DONE. We agree this is confusing and have removed the sentence.
The tendency for smaller abnormal larvae to survive longer and at L 156 79% abnormal in day 18 – prompts the question was this due to mass cloning? With what we know about echinoderm larval biology that seems very possible and should be acknowledged. A paper on starfish larval cloning (# 46) that shows food levels influence the incidence of cloning is cited but cloning is not mentioned, please expand.	DONE. A pertinent point. Thank you. We have added text (Ln. 302-307) to describe how our results could have been due to cloning.
L. 185 188 add x-x settled larvae per replicate tube – or in the figure legends give the range in the number of larvae in the tubes. Figure 5 is very interesting.	DONE. The numbers of larvae in replicates added (Ln. 218 and 222).
Reviewer #3	Author response to reviewer:
Dear authors and editor, I find that my comments have been met adequately. I have no further comments.	We thank Reviewer 3 for their helpful suggestions.